# Hyper-Kamiokande †

## Michael B. Smy [1,2] on behalf of the Hyper-Kamiokande Collaboration

1    Department of Physics and Astronomy, University of California, Irvine, 3177 Frederick Reines Hall, Irvine, CA 92697, USA; msmy@uci.edu; Tel.: +1-949-824-7502

2    Kavli Institute for the Physics and Mathematics of the Universe, The University of Tokyo Institute for Advanced Study, University of Tokyo, 5-1-5 Kashiwanoha, Kashiwa, Chiba 277-8583, Japan

†    Presented at the 23rd International Workshop on Neutrinos from Accelerators, Salt Lake City, UT, USA, 30–31 July 2022.

**Abstract:** Hyper-Kamiokande, featuring a 260 kton cylindrical water Cherenkov detector, is one of the defining next-generation neutrino experiments. In addition to investigating neutrino oscillations with a dedicated, high-intensity muon neutrino beam, it will study atmospheric neutrinos, solar neutrinos, supernova neutrinos, and other astrophysical neutrinos. Its physics sensitivity is similar but complementary to other next-generation neutrino experiments: DUNE, a 40 kton liquid Argon time projection chamber, and JUNO, a 20 kton liquid scintillator detector.

**Keywords:** neutrino oscillations; CP violation; astrophysics; supernova





## 1. Introduction

Hyper-Kamiokande [1] (HK) is the third instance in a series of water Cherenkov detectors in Japan: the 3 kton Kamiokande experiment searched for nucleon decay, observed the directional distribution of neutrinos originating from the sun [2], detected supernova neutrinos from SN1987a in the large Magellanic cloud, and saw a hint of atmospheric neutrino oscillations. The 50 kton Super-Kamiokande established atmospheric and solar neutrino oscillations, continued the nucleon decay search with an improved sensitivity, and studied neutrino oscillations with two neutrino beams produced by proton accelerators (K2K and T2K) confirming the dominant $\nu_\mu \to \nu_\tau$ oscillation and establishing a "subdominant" oscillation mode $\nu_\mu \to \nu_e$, as well as finding a hint of leptonic CP violation. The basic layout of all three detectors is the same: a cylindrical volume of water surrounded by photodetectors arranged on a grid. Hyper-Kamiokande uses two newly developed photosensors: "box and line" photomultiplier tubes (PMTs) of 20″ diameter (see Figure 1) and clusters of 3″ PMTs (mPMTs). The detector design includes 20,000 20″ PMTs and the full number of mPMT modules is not yet fixed. Events are triggered by PMT coincidence within a time window of a few 100 ns. The interaction vertex is reconstructed from the timing of the PMT signal and the light pattern decides the number of tracks and particle ID (showering or "electron"-like vs. track or "muon"-like). Energy reconstruction relies on the total amount of detected light. Due to its large size, HK will measure beam neutrinos, atmospheric neutrinos, solar neutrinos, and supernova neutrinos with a very high statistical accuracy. To make full use of this, the HK project includes upgrades to T2K near detectors, which monitor and characterize neutrino beams. In particular, a new "intermediate" water Cherenkov detector (IWCD) will be built. It will measure cross-sections in the water using the Cherenkov technique (with mPMTs) and study them as a function of the off-axis beam angle. This angle is correlated with the beam peak energy, so the neutrino energy need not be inferred from the visible energy in the detector. This is crucial to reducing systematic uncertainties due to cross-section modeling. To control the off-axis beam angle, the detector will be moved vertically (and of course the reconstructed neutrino interaction vertical position will be used).

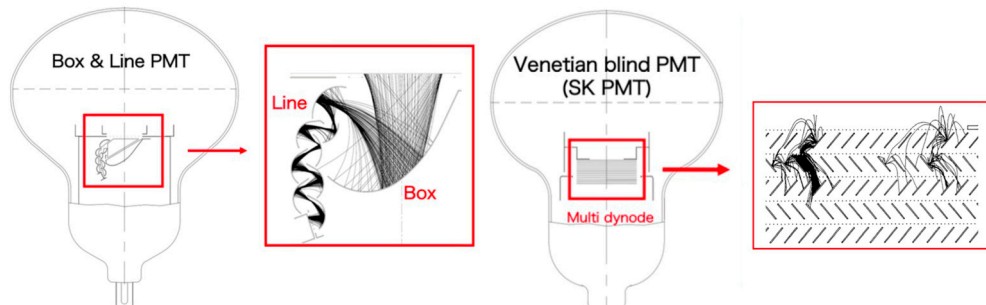

**Figure 1.** "Box and line" (**left**) and "Venetian blinds" (**right**) PMT [3].

## 2. Photomultiplier Tubes and In-Water Electronics

The newly developed "box and line" PMTs have the same size (20″ diameter) as the "venetian blinds" PMTs used in the Super-Kamiokande detector, but a higher quantum efficiency and different dynode structure. The transit time spread (2.7 ns) of the box and line PMT is smaller (leading to better time resolution at one photo-electron equivalent pulse height) and they also count photo-electrons much better. Each PMT covers about 2000 cm$^2$, and the light yield of 20,000 installed 20″ PMTs is about 6/MeV. The dark noise is about 4 kHz. The mPMT modules consist of 19 3″ PMTs arranged in 20″ diameter housing. The transit time spread is smaller than the 20″ PMTs (1.3 ns). Each module covers about 870 cm$^2$. The light yield of 5000 installed modules is about 1/MeV. The 3″ PMT dark noise rate is 200–300 Hz/PMT. The better granularity and directional sensitivity (the 3″ PMTs are not arranged facing parallel, but point in slightly different directions) will improve the detector systematics and energy calibration. Like Super-Kamiokande, there is a (very thin) outer detector surrounding the inner detector to veto entering events such as cosmic ray muons. It is read out by 3″ PMTs with wavelength shifter plates. Unlike Super-Kamiokande, the electronics for both the inner and outer detector are situated in water-proof containers to minimize the cable lengths for the PMTs. As in Super-Kamiokande, the outer detector will be covered in reflective Tyvek to improve the effective light yield. The optical barrier to the inner detector consists of black plastic ("black sheet"), also similar to the Super-Kamiokande design.

## 3. Neutrino Oscillations with the Neutrino Beam and Complementarity with DUNE and JUNO

Standard neutrino oscillations are described by a unitary effective flavor mixing matrix (parametrized by three mixing angles $\theta_{12}$, $\theta_{13}$, and $\theta_{23}$, and a CP-violating phase $\delta$), as well as three mass$^2$ splittings, $\Delta m^2_{21}$, $\Delta m^2_{31}/\Delta m^2_{32}$, which lead to relative phase shifts of the three neutrino wave functions as they propagate, resulting in oscillations of the neutrino flavor. The interactions of the neutrinos with the matter density they pass through also affect these phase shifts and therefore the neutrino oscillations. This "matter effect" depends on the neutrino mass ordering (MO) and has the opposite sign for anti-neutrinos: matter effects therefore generate a CP asymmetry of neutrino oscillations, just as $\delta$ does. The high-intensity neutrino beam from Tokai gives excellent sensitivity for testing and studying all aspects of neutrino oscillations, but in particular CP violation due to $\delta \neq 0, \pi$. Figure 2 gives the expected significance of the CP discovery as a function of the true phase $\delta$. Due to the short (about 300 km) baseline, the matter effect sensitivity and, consequently, the MO sensitivity are smaller compared to DUNE, but this also makes it harder for DUNE to disentangle earth matter effects from $\delta$ (and the octant of $\theta_{23}$). Additionally, the shortness of the baseline allows for a lower energy beam (at the first oscillation maximum): like T2K (and unlike DUNE), HK uses a narrow-band beam of about 600 MeV. While all three major cross-section channels (quasi-elastic, resonant pion production, and deep inelastic) contribute significantly to the charged-current interactions of the DUNE neutrino beam, for HK, the quasi-elastic cross-section dominates. This is of great benefit to a water Cherenkov detector with limited tracking abilities. HK also has good sensitivity to $\theta_{23}$ and its octant:

with improved systematics (compared to T2K 2018) the wrong octant will be excluded at >3σ in the range of $0.47 < \sin^2\theta_{23} < 0.55$. The combination of HK and DUNE oscillation measurements will benefit from different sensitivities to cross-section systematic effects; this, in turn, leads to a more robust and precise measurement of the phase δ and other oscillation parameters. JUNO measures the low-energy reactor neutrino oscillation where the cross-section (inverse beta decay) is known with good precision; the MO is determined by measuring all three mass$^2$ splittings in the electron-flavor disappearance channel, but there is no sensitivity to δ (or the $\theta_{23}$ octant).

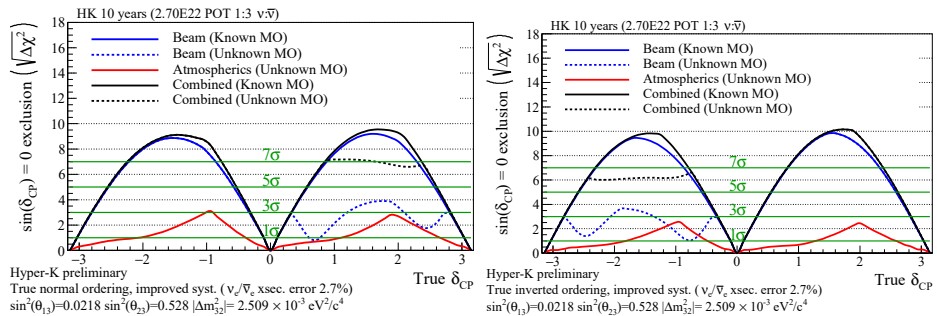

**Figure 2.** CP violation for known and unknown, normal and inverted mass ordering from beam neutrinos and the combination of beam and atmospheric neutrinos.

Agarwalla, Das, Giarnetti, Meloni and Singh [4] compared the sensitivities of the combination of HK and DUNE oscillation data vs. just HK or DUNE by themselves for the "Snowmass meeting": when considering a CP sensitivity benchmark of establishing CP violation with at least a 3σ significance for 75% of all the true δ values (independently from all the other oscillation parameters), they find that both experiments fall a bit short: DUNE suffers from a $\theta_{23}$-δ degeneracy, and HK needs tighter systematic uncertainties than its current goal. However, combining both data sets, this "75% coverage" could be met with even half of the planned neutrino exposure. In the case of DUNE, even doubling its exposure would not resolve the $\theta_{23}$-δ degeneracy, while a HK with reduced systematic errors (2.7%) would get close to 75% coverage. Alternatively, an additional far detector in Korea would also meet this standard, but the coverage would always be less than that of HK + DUNE. Furthermore, with tight systematic errors, δ is determined with a better than 10-degree accuracy for all the true values of δ (at maximal mixing $\sin^2\theta_{23} = 0.5$), if data from both experiments are combined.

## 4. Neutrino Oscillations with Atmospheric Neutrinos

Atmospheric neutrino oscillations are useful for addressing the reduced HK beam neutrino sensitivity to matter effects and MO: atmospheric neutrinos have baselines up to 12,000 km (determined by the neutrino's zenith angle), so the matter effects are much stronger, and even resonant oscillations occur at some zenith angles and energies (see Figure 3). If the mass ordering is not known from previous experiments, the HK beam neutrino CP violation sensitivity is also impacted by the resulting ambiguity (see Figure 2). However, the large size of the HK detector allows for the subdivision of the atmospheric neutrino data sample into many bins in reconstructed energy and zenith angles to study these "resonant regions" of the Earth where the oscillation probability is enhanced. While the resolution in the neutrino zenith angle and energy reconstruction smears out the impact of these resonances somewhat, HK atmospheric neutrino interactions can still select the correct mass orderings with a 3–5σ significance within ten years (see Figure 3). As shown in Figure 2, the combined neutrino oscillation analysis of HK atmospheric and beam neutrinos recovers much of the CP sensitivity in the case of unknown mass ordering, as the longer baseline of the atmospheric neutrinos assists in selecting the MO and resolving the ambiguities of the beam-only measurement. In addition, atmospheric neutrinos help to constrain systematic uncertainties (detector effects, as well as cross-section-related

systematics), and their wide energy range make them the perfect tool to search for more exotic (neutrino flavor) physics, such as Lorentz invariance violation [5] or non-standard neutrino interactions [6].

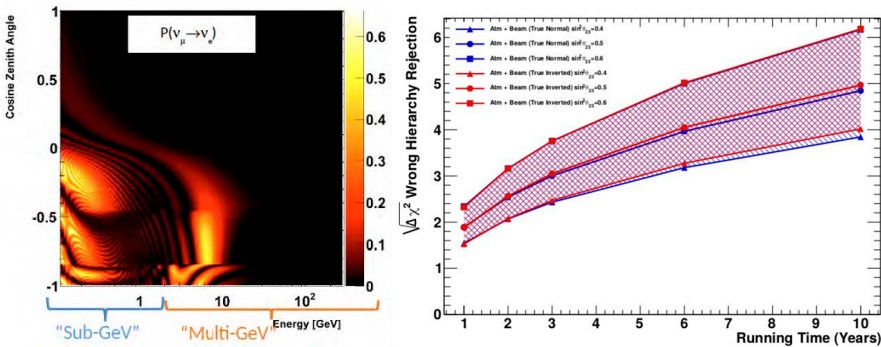

**Figure 3.** (**Left**): Atmospheric neutrino muon to electron flavor oscillation probability as a function of zenith angle and energy (normal ordering). (**Right**): Expected significance of the wrong mass ordering rejection as a function of HK running time.

## 5. Nucleon Decay

The core prediction of grand unified theories is nucleon decay, the original motivation for the first Kamiokande detector. Today, SK data provide the most stringent limits on almost every single nucleon decay mode. Many modes are still not background-dominated. Therefore, the larger mass of HK further probes these grand unified theories, e.g., $p \rightarrow e^+\pi^0$ will be tested up to a lifetime of $10^{35}$ years. In general, water Cherenkov detectors have a good sensitivity, since the oxygen nucleus is quite light and 10% of the protons are even free (all the neutrons are bound). Therefore, produced pions can fairly easily escape the nucleus. On the other hand, the charged mesons produced in nucleon decays (e.g. in the mode $p \rightarrow K^+\nu$) are often near or even below the Cherenkov threshold, which degrades the sensitivity to such modes in Water-Cherenkov detectors. For such decay modes, DUNE and JUNO have a better sensitivity per kton, but due to the large size of HK, it will remain competitive even for these modes. Figure 4 shows the expected sensitivity for many nucleon decay modes, as well as $p \rightarrow e^+\pi^0$ as the representative of an "easy" mode and $p \rightarrow K^+\nu$ as the representative of a more challenging mode.

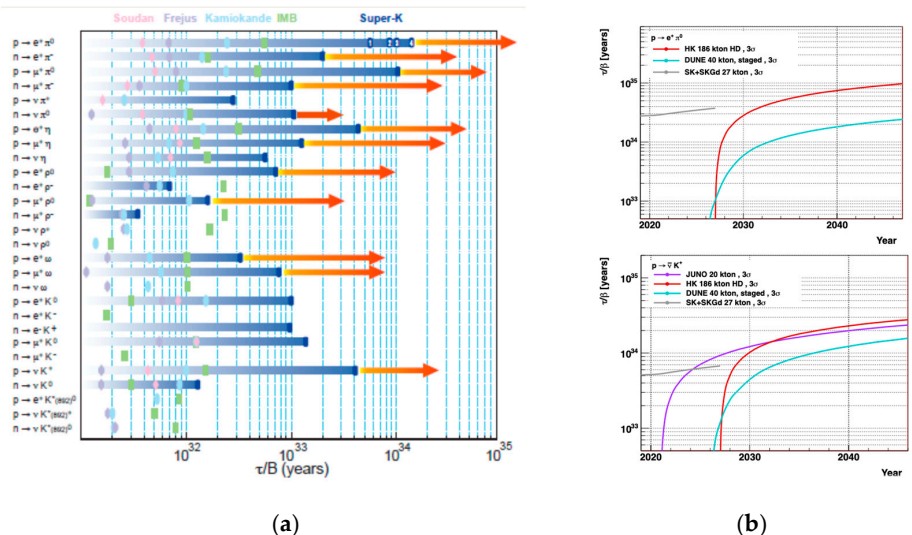

(**a**)      (**b**)

**Figure 4.** HK nucleon decay sensitivity: (**a**) Expected 90% C.L. limits for many modes in HK compared to Frejus, Kamiokande, IMB, SK, and HK. (**b**) Expected 90% C.L. limits for $p \rightarrow e^+\pi^0$ and $p \rightarrow K^+\nu$ in HK compared to SK, JUNO, and DUNE.

## 6. Neutrinos from Core-Collapse Supernovae

HK's huge size opens up the possibility of observing the neutrino bursts from core-collapse supernovae in M31 at a distance of 765 kpc: 6–10 events are expected. This approximately doubles the chances of core-collapse supernovae with observable neutrino bursts (compared to just milky way supernovae) during HK's runtime. In total, 50,000 to 80,000 neutrino interactions are expected from a galactic supernova (at the center). Such a high-statistics burst allows for detailed studies of the neutrino and astrophysical properties (e.g., explosion mechanism, proto-neutron star formation, and black hole formation). Even with just a few 100 events [7], HK can distinguish five of the most recent supernova models from each other with a 97% probability, so a supernova at a distance of ~50 kpc (such as SN1987a) with 1200 to 2000 events would be quite enough to select the best model. The supernova neutrino burst detection at HK is also quite valuable for multi-messenger astronomy, since HK can identify the supernova direction, and the neutrino signal arrives up to a few hours earlier than the optical one. The main supernova neutrino sensitivity for HK is in the electron anti-neutrino channel, but electron neutrino sensitivity exists via electron-neutrino elastic scattering. This cross-section is considerably smaller, but the initial neutronization burst (which is only $\nu_e$) is still observable for a good fraction of the galaxy. The joint observation of a supernova neutrino burst by HK and DUNE is particularly exciting, as DUNE's sensitivity is primarily in the electron neutrino channel. "Collective effects" (a kind of neutrino oscillation matter effect from the high neutrino density in an exploding star) may show up as non-thermal spectral features in either the electron neutrino or anti-neutrino flux, depending on the mass ordering being normal or inverted. This means that one experiment would see non-thermal features and the other would not, providing strong evidence for these "neutrino–neutrino" interactions and selecting the mass ordering at the same time.

In addition, HK is sensitive to the diffuse neutrino flux from all supernovas [8] up to a redshift of about 1 (beyond that, the neutrino spectrum gets too soft to be distinguished from the reactor neutrinos produced on Earth). Here, the anti-neutrino channel is an advantage, as there is no significant solar anti-neutrino background; in the neutrino channel (the main channel for DUNE), searches become sensitive only above ~19 MeV (the solar *hep* neutrino endpoint), unless the neutrino candidate's direction can be obtained. For diffuse supernova neutrino detection, a high photo-cathode coverage of HK becomes critical: without doping with Gd ions, the search can only identify inverse-beta decays via neutron captures on hydrogen (2.2 MeV $\gamma$ emission), or must rely solely on the positron for inverse beta decay detection: in ten years, the expected signal significance is about 4$\sigma$ (3$\sigma$) for a 40% (30%) photocoverage based on $70 \pm 17$ ($40 \pm 13$) events.

## 7. Solar Neutrinos

Kamiokande pioneered solar neutrino observation with the water Cherenkov technique using electron-neutrino elastic scattering [2]. Its sensitivity is limited to high-energy solar neutrinos ($^8$B and *hep*) with comparably small fluxes. In particular, HK's large size might allow for the detection of the lowest solar neutrino flux, the *hep* neutrinos. The photo-cathode coverage and resulting light yield strongly affect this sensitivity, both due to the recoil electron energy threshold and the energy (and vertex) resolution. The energy resolution is important for both radioactive background rejection (which is the main impact of vertex resolution), as well as the discrimination of *hep* solar neutrinos from more abundant $^8$B neutrinos. Figure 5 shows the expected spectrum with a 40% photocathode coverage assumed. After ten years, the *hep* component can be separated with a 2–3$\sigma$ significance. HK will monitor the solar core in real time: the interaction rate of the $^8$B neutrinos will be of the order of 5/hour (if the efficiencies are similar to those in Super-K). The inferred $^8$B flux is very sensitive to the solar core temperature.

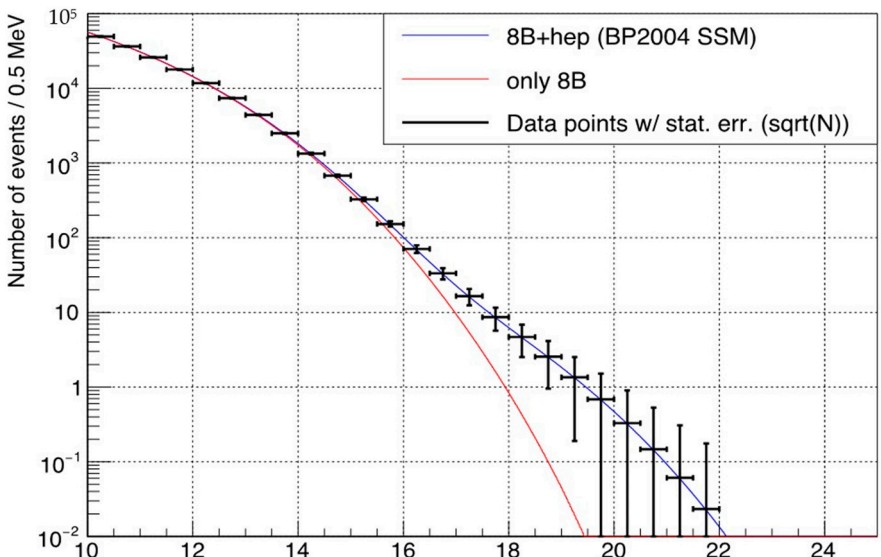

**Figure 5.** Expected HK recoil electron spectrum from solar neutrino-electron elastic scattering. Photocathode coverage is assumed to be 40%. The dominant $^8$B component (red) has a smaller endpoint than *hep* (the blue line shows $^8$B + hep combined).

In HK, an additional challenge for solar neutrino analyses is HK's lack of depth, resulting in a much larger cosmogenic radioactive background. New techniques to tag such backgrounds were recently developed at Super-K [9], in particular hadronic shower tagging and reconstruction. Hadronic showers were directly tagged and reconstructed from the "cloud" of neutrons produced by these showers. In Super-K, ~50% of the radioactive decays of the spallation products were preceded by neutron clouds near the vertex of the decay events (see Figure 6), with the neutron detection signal being neutron capture on hydrogen (and the subsequent release of a single 2.2 MeV $\gamma$). In the HK baseline design, no Gd is added to the water, so the detection would happen in a similar fashion (in Super-K, the efficiency of the neutron cloud tagging method was improved with the addition of $Gd_2(SO_4)_3$; neutron captures on Gd release eight MeV $\gamma$ cascades). The second method tagged showers by observing multiple spallation product decays originating from the same shower. Both methods led to only a minimal neutrino exposure loss for decay times of up to 60 s. The third method reconstructed showers (electromagnetic or hadronic) along muon tracks by looking for a peak in the "optical dE/dx" of the track. The longitudinal (difference of the dE/dx peak and closest point of the muon track to the observed decay position) and transverse distance of a candidate event to the muon track were combined with the muon observed charge and the time difference to form a likelihood. All three methods combined resulted in 10% more solar neutrino interactions, without increasing spallation-related backgrounds (see Figure 6). These methods will be critical to obtaining precise solar neutrino results in HK; without them, the loss of neutrino exposure due to a higher spallation rate will reduce the statistical advantage of HK over Super-K.

The main particle physics interest is the study of matter effects on solar neutrino flavor conversion: the Mikheyev–Smirnov–Wolfenstein (MSW) effect [10,11] adiabatically converts electron-flavor neutrinos into the second-mass eigenstate if the neutrino energy is above O(10 MeV). The resulting "solar $\nu_2$ beam" is used to test the terrestrial matter effects by measuring the electron-flavor content without (during the day) and with (during the night) earth matter. The "day/night asymmetry" is defined as $A_{DN} = (D - N)/(0.5(D + N))$, if D (N) denotes the neutrino interaction rate during the day (night). Terrestrial matter effects result in a negative day/night asymmetry due to electron flavor regeneration. For the elastic scattering of $^8$B neutrinos on electrons, this asymmetry is, at most, a few % [12]. The magnitude of $A_{DN}$ depends mostly on $\Delta m^2_{21}$. Figure 7 shows the sensitivity of HK to measuring a non-zero $A_{DN}$ and distinguishing the solar neutrino best-fit prediction of $A_{DN}$

from predictions based on KamLAND's $\Delta m^2_{21}$ measurement. For energies below ~5 MeV, the MSW effect transitions to average vacuum oscillations, leading to a larger electron-flavor content. Figure 7 shows the HK sensitivity to the resulting spectral distortions.

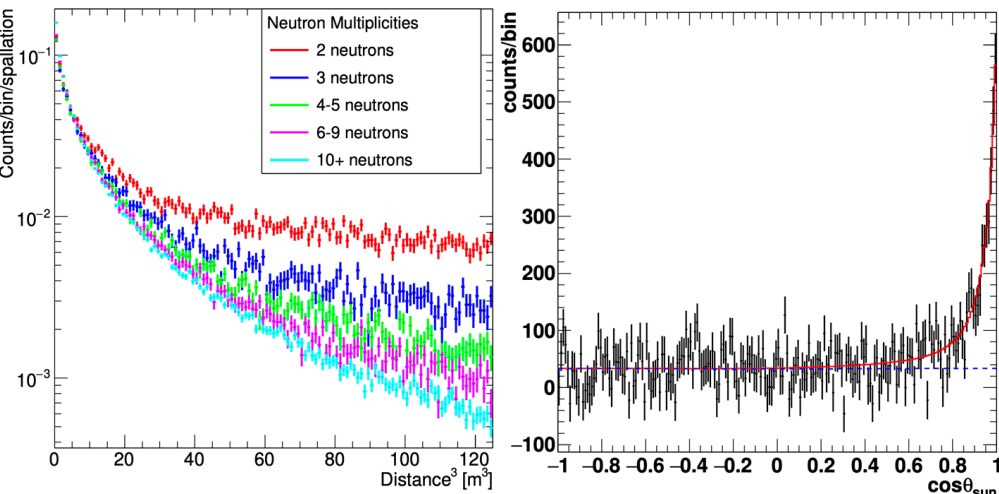

**Figure 6. Left**: Distance³ of the "center of gravity" of "neutron clouds" from hadronic showers with the observed radioactive decays. **Right**: Angular distribution of additional solar neutrino recoil electron candidates with respect to the solar direction after employing new tagging methods.

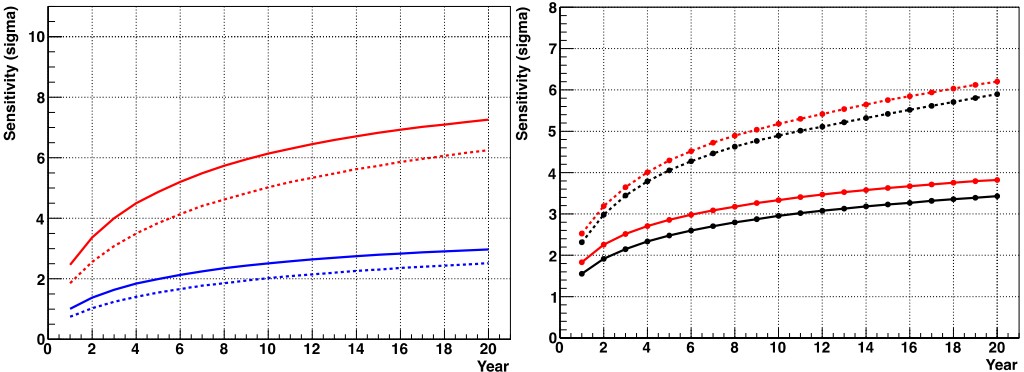

**Figure 7. Left**: Sensitivity of HK to find a non-zero day/night asymmetry (red) and separate predictions based on the solar and KamLAND best-fit $\Delta m^2_{21}$ values. Solid lines are for 40% photo-cathode coverage and dotted lines for 20%. **Right**: Sensitivity of HK to find significant spectral distortions due to the MSW effect. The solid (dashed) lines are for a 3.5 MeV (4.5 MeV) threshold of recoil electron kinetic energy. The black (red) line assumes the best solar neutrino $\Delta m^2_{21}$ fit of 2019 (2020).

## 8. Summary

Hyper-Kamiokande will be a multi-purpose experiment studying neutrino oscillations using neutrino beams, as well as detecting "natural sources" such as atmospheric, solar, and supernova neutrinos. Its other physics topics are nucleon decay searches, indirect dark matter searches, and multi-messenger astronomy. As Hyper-Kamiokande measurements are highly complementary to DUNE's, the combination of both experimental programs is much more sensitive than DUNE or HK by themselves.

**Funding:** This presentation (as well as Super-Kamiokande related research) was supported by the National Science Foundation (NSF) Award Number PHY-2013073.

**Institutional Review Board Statement:** Not applicable.

**Informed Consent Statement:** Not applicable.

**Data Availability Statement:** Not applicable.

**Conflicts of Interest:** The author is affiliated with UCI and Kavli IPMU and the presentation was supported by NSF grant PHY-2013073.

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
