# Peer review of "Hyper-Kamiokande†"

_psf, doi:10.3390/psf8010041_

Round 1

Author Response

abstract: 260 kton. changed as suggested.

page 2: with the matter they pass -> with the matter density they pass through

page 2: 1.3 ns: changed as suggested

page 2: the three mass square differences: changed as suggested, the referee is correct that only two are independent

page 2: JUNO's -> JUNO (changed as suggested)

page 2: known to -> known with

reference 3: added M. Singh, my apologies for this omission

Reviewer 2 Report

Hyper-Kamiokande is an important experiment in the field of neutrinos today. This paper is a good introduction to the importance of the Hyper-Kamiokande experiment. What's more valuable is that the paper shows the complementarity between Hyper-Kamiokande and other experiments. Worth publishing and being read. 

There are a few places I would like to improve.

1. For "box and line" PMTs, it is best to add a figure to show the appearance of the PMT.

2. Figure 1. The figure on the right and the figure on the left are repeated.

3. The text on the right of Figure 2. and the left of Figure 3. is unclear. Change to a clearer figure or enlarge the figure.

4. 40% photocathode coverage already determined? Because a lot of data in the paper is based on this coverage. Please confirm that this coverage has been established or is the minimum coverage for future experiments.

5. The first cited reference has no content.

Author Response

The reviewer suggests five improvements. Here are my replies:

(1) I follow the reviewers suggestion to add a figure for the box and line PMT. The figure compares the box and line to the Venetian blind design.

(2) The reviewer notes that the left and right panel of Figure 1 are identical. I apologise for the mistake. It is corrected now.

(3) The reviewer believes that the text of Figure 2 and 3 are unclear. It seems to me that both texts are readable. The size of these Figures is limited by the imposed page limit.

(4) The HK baseline design is now 20% photocathode coverage. Originally, 40% was envisioned. Some solar neutrino physics depends significantly on the photocathode coverage. In some cases (Fig. 6), the paper compares both coverages, but it is difficult to do so for the 8B/hep neutrino separation due to the page limit. However, this is determined by energy resolution which is limited by photon counting statistics (and therefore differs by sort(2) when going from 40% to 20% coverage) so the expected degradation is fairly easy to understand. Some of the solar analysis techniques discussed in the paper were developed with 40% coverage in Super-Kamiokande. They may work less well in a larger detector, and with smaller photocathode coverage. Since the discussion in the paper is based on Super-Kamiokande data, rather than Hyper-Kamiokande Monte Carlo simulations, there is no 20% case discussed.

5. The referee states the first reference has no content. I checked and found the first reference easily (arXiv:1805.04163). It takes a while to download the pdf as it has many pages.